# Evaluation of Equivalent Flexural Strength for Complete Removable Dentures Made of Zirconia-Impregnated PMMA Nanocomposites

**DOI:** 10.3390/ma13112580

**Published:** 2020-06-05

**Authors:** Saleh Zidan, Nikolaos Silikas, Julfikar Haider, Abdulaziz Alhotan, Javad Jahantigh, Julian Yates

**Affiliations:** 1Dentistry, School of Medical Sciences, University of Manchester, Manchester M13 9PL, UK; nikolaos.silikas@manchester.ac.uk (N.S.); abdulaziz.alhotan@postgrad.manchester.ac.uk (A.A.); javad.jahantigh@mft.nhs.uk (J.J.); julian.yates@manchester.ac.uk (J.Y.); 2Department of Dental Materials, Faculty of Dentistry, Sebha University, Sebha, Libya; 3Department of Engineering, Manchester Metropolitan University, Manchester M1 5GD UK; j.haider@mmu.ac.uk

**Keywords:** denture base, high-impact PMMA, zirconia (ZrO_2_), nanocomposite, flexural strength, fatigue loading

## Abstract

High-impact (HI) polymethyl methacrylate (PMMA), obtained from modification of conventional PMMA, is commonly used in prosthodontics as a denture base material for improved impact resistance. However, it suffers from poor flexural strength properties. The aim of this study was to investigate the flexural strength of complete removable dentures made of HI heat-polymerised PMMA resin reinforced with zirconia nanoparticles at two different concentrations. The effect of fatigue loading on the flexural strength behaviour of the dentures was also investigated. A total of 30 denture specimens were fabricated from PMMA with different concentrations of zirconia nanoparticles: 0 (control), 3, and 5 wt.%. Ten specimens in each group were divided into two subgroups, with five specimens in each, to conduct both flexural strength and fatigue loading test of each of the subgroups. Fatigue loading was applied on the dentures using a mastication simulator and equivalent flexural strength was calculated with data from bending tests with and without fatigue cyclic loading. One-way analysis of variance (ANOVA) of the test data was conducted with the Bonferroni significant difference post-hoc test at a preset alpha value of 0.05. Paired *t*-test was employed to identify any difference between the specimens with and without the application of fatigue loading. The fractured surface of the denture specimens was examined with a scanning electron microscope (SEM). The bending tests demonstrated that the mean equivalent flexural strength of reinforced HI PMMA denture specimens with 5 wt.% zirconia nanoparticles increased significantly (134.9 ± 13.9 MPa) compared to the control group (0 wt.%) (106.3 ± 21.3 MPa) without any fatigue loading. The mean strength of the dentures with PMMA +3 wt.% zirconia also increased, but not significantly. Although the mean strength of all specimen groups subjected to fatigue loading slightly decreased compared to that of the specimen groups without any fatigue cyclic loading, this was not statistically significant. Denture specimens made of HI heat-polymerised PMMA reinforced with 5 wt.% zirconia nanoparticles had significantly improved equivalent flexural strength compared to that made of pure PMMA when the specimens were not subjected to any prior fatigue cyclic loading. In addition, the application of fatigue cyclic loading did not significantly improve the equivalent flexural strengths of all denture specimen groups. Within the limitations of this study, it can be concluded that the use of zirconia-impregnated PMMA in the manufacture of dentures does not result in any significant improvement for clinical application.

## 1. Introduction

The acrylic resin polymethyl methacrylate (PMMA) is widely used for the manufacture of dental prostheses, including conventional removable complete or partial dentures and implant-supported prostheses [1]. Acrylic resins have many advantages, including acceptable aesthetic appearance, lightness, biocompatibility, and ease of processing in the laboratory for clinical use [2,3]. However, this material is still some way from possessing the ideal mechanical properties for denture base and other prosthetic applications, suffering from low resistance to impact, flexural weakness, and fatigue [4]. The fracture of dentures is the most frequent problem when patients present with failures of their prostheses. Many of these fractures occur inside the mouth as a result of denture fatigue caused by mastication processes [5]. It is widely accepted that many materials suffer a loss of strength as a result of cyclical stress over a long period of time. Microcracks start to generate at the point of alternating stresses in the denture, propagate through the material, and finally lead to fatigue failure after a certain period of time [3,6]. Flexural fatigue of PMMA has been determined as a cause of midline fractures of complete dentures [7]. Majority of fractures occurred in the midline of maxillary complete dentures, with the incidence being 2–3 times higher when compared to mandibular dentures [8,9]. Additionally, acrylic resin dentures demonstrated flexing during functioning to a much greater degree than expected, as well as poor tissue adaptation [6]. Various attempts have been made in the past to improve the mechanical properties of denture base acrylic resins by incorporating particles, wires, fibres, or mesh aligned with the shape of the denture base [10,11,12,13,14]. One of the most notable developments was based on the chemical modification of conventional PMMA with rubber particles (butadiene-styrene) with sizes ranging from 1 to 5 µm, marketed as a “high-impact” variation [14]. This has been successful, to a certain extent, in improving the impact strength and dimensional stability [15,16,17,18]. However, the incorporation of rubber decreases the flexural and fatigue strengths and the modulus of elasticity compared to conventional heat-polymerised acrylic resins [15,17,18,19]. Several in vitro studies have investigated performance of metal wire-reinforced acrylic dentures [10,11,13]. Maxillary complete dentures made of acrylic resin reinforced with metal wire (Cr–Co alloys) were placed under the ridge lap in the anterior region and in the anterior and posterior regions of the denture base. The tests showed that the wire reinforcement increased the flexural strength of the dentures [13]. However, reinforcement of acrylic resin with metal wires often resulted in wire separation at the interface due to poor adhesion between the denture base resin and metal reinforcement [20,21]. In addition, the addition of metal wire often resulted in unacceptable denture aesthetics and significantly increased the overall mass of the denture base [22].

Exploring substitutes for metal reinforcement, other studies have reported incorporation of microfibres such as aramid, ultrahigh molecular weight polyethylene, carbon, nylon, urethane oligomer and E-glass in the forms of chopped, flaked, continuous, or woven constituents [3,11,23]. Several studies concluded that glass-fibre reinforcement significantly increased flexural strength, flexural modulus, and impact strength of acrylic dentures [3,7,10,11,13,20,23]. Vallittu et al. and Im et al. reported improved fatigue and fracture resistance from a complete denture made of acrylic resin reinforced with glass fibres with force magnitudes of 80, 100, and 180 N applied to the occlusal surfaces of the test specimens with fatigue cyclic loading repeated at 300,000 masticatory cycles in a mastication simulator machine, the equivalent to real-life use for over a year [7,10,12]. A clinical study investigated the orientation of the glass fibres in PMMA dentures and suggested that they should be placed close to the location of highest tensile stress to prevent any initiation of fracture. However, incorrect positioning of glass fibres could lead to a decrease in mechanical properties [13,24]. Therefore, it is challenging to manufacture dentures with accurate fibre positioning and to maintain the mechanical properties consistently from a quality control point of view.

Recently, numerous investigations have focused on adding nanoparticles such as yttria-stabilised tetragonal zirconia polycrystals (Y-TZP) to improve the mechanical and physical properties of conventional heat-polymerised denture base resins [25,26,27]. This type of zirconia, called “ceramic steel”, possesses superior mechanical properties, good surface properties, and high biocompatibility, thus making it an attractive option for many dental applications [28]. In the authors’ previous study [26] with beam-type PMMA–zirconia nanocomposite samples, it was determined that optimum flexural strength can be obtained by adding zirconia nanoparticles at approximately 3 and 5 wt.%.

To date, the effect of zirconia nanoparticles on the flexural strength and fatigue loading cycles of HI PMMA has not been evaluated with specimens of a similar shape to a complete denture. Therefore, the aim of this study was to investigate the flexural strength properties of complete dentures made from HI heat-polymerised PMMA resin reinforced with zirconia nanoparticles with and without fatigue cyclic loading in a mastication simulator. The hypothesis was that HI heat-polymerised PMMA incorporated with zirconia nanoparticles with and without fatigue cyclic loading would lead to a significant increase in value of the flexural strength of the complete dentures.

## 2. Materials and Experimental Method

### 2.1. Materials

A commercially available Metrocryl HI denture base powder, PMMA (polymethyl methacrylate), and Metrocryl HI (X-Linked) denture base liquid (MMA, methyl methacrylate) (Metrodent Limited, Huddersfield, UK) were selected as the denture base material. Yttria-stabilised zirconia (ZrO_2_) (94% purity; Sky Spring Nano Materials, Inc., Houston, TX, USA) nanoparticles with an average size between 30 and 100 nm were chosen as the inorganic filler agent for fabricating the nanocomposite denture specimens, as shown in Table 1.

### 2.2. Selection of Appropriate Percentages of Zirconia Nanoparticles

Three groups of complete dentures were prepared for this study by the first researcher and their compositions are described in Table 2. All had an acrylic resin powder-to-monomer ratio of 21 g:10 mL, in accordance with the manufacturer’s instructions. The particle salinisation procedure can be found in [26].

### 2.3. Preparation of Complete Removable Dentures

Maxillary edentulous master casts were duplicated using an addition cure silicone putty to obtain a mould that was then used to produce thirty edentulous casts by pouring high-strength dental stone into the silicone mould. Two sheets of baseplate wax (Metro wax, Metrodent Limited, Huddersfield, UK) with a thickness of 3.50 mm were adapted onto the palatal surface on the edentulous cast, and then the occlusal rim was placed on denture base wax. The maxillary master cast with occlusal rim was fixed on an articulator (John Winter, Halifax, UK) using dental plaster in preparation for setting the teeth. Then, the maxillary anterior teeth (Artic 6M S10 shade BL3, Metrodent Limited, Huddersfield, UK) and maxillary posterior teeth (Artic 8M 10 30U shade A2, Metrodent Limited, Huddersfield, UK) were fixed onto the occlusal rim. Upon completing waxing of the denture base and teeth, the wax denture and plaster cast were removed from the articulator and placed in a flask filled with dental plaster and dental stones were place onto the teeth, after setting. The denture wax was then removed through dewaxing process.

The silane-treated zirconia and acrylic resin powders were weighed according to Table 2, using an electronic balance with an accuracy of three decimal points (Ohaus Analytical plus, Ohaus Corporation, Parsippany, NJ, USA). Where indicated, zirconia powder was added at the appropriate concentration to the acrylic resin monomer and mixed in a speed mixer (DAC 150.1 FVZK, High Wycombe, UK) at 2500 rpm for 5 min. Once mixed, the acrylic resin powder was then added to the solution, and mixed again in accordance with the manufacturer’s instruction, until a smooth, uniform mixture was obtained. The mixture was then packed into the flask, pressurised, and immersed in a curing water bath for 6 h to allow polymerisation. 

The flask was then removed from the curing bath and left to cool 30 min at room temperature. The flask was then opened, and the denture removed. The denture was placed in an ultrasonic cleaning machine containing water (Elma Electronic, Bedford, UK) to remove any attached stone, trimmed using a tungsten carbide bur (D B Orthodontics, Yorkshire, UK), ground with an emery paper grit 40 (Norton, Saint-Gobain, Stafford, UK) and, finally, polished with pumice powder in a polishing machine (Tavom, Wigan, UK). All thirty denture specimens were fabricated individually in the manner detailed above.

### 2.4. Mechanical Strength Test

Fatigue cyclic loading was performed according to a previous study by Im et al. using a chewing simulator (CS-44.2 SD Mechatronik GmbH, Westerham, Germany) that simulated maxillary and mandibular movement in the mouth during mastication (Figure 1) [10].

Fifteen denture specimens, five from each group, were employed for fatigue tests. The upper bar faced the centre of midline of the denture and T-shaped jig was placed against the second premolar (full) and first molar (partial) teeth on each side at 1.6 Hz. The initial loading applied on the denture was 8 kg (78.48 N). The vertical movement of the T-bar was set at 2 mm and lateral movement at 0.2 mm, with a vertical speed of 30 mm/s. A total number of 250,000 mastication cycles were performed in 37 °C distilled water to simulate approximately one year of use in an oral environment.

Thirty denture specimens with and without fatigue cyclic loading were subjected to three-point bending test in a Hounsfield universal testing machine (Hounsfield Tensometer, H10KS, Birmingham, UK). The distance between the second molar teeth (last tooth on each side) acted as a supporting span with a length of 42.33 ± 0.2 mm. The load was applied to the palatal fitting surface at a crossover point between the palatal midline and the line connecting the centre of first molars on each side of the denture.

The thickness of the dentures was measured using a digital micrometer (Mitutoyo, Andover, UK) at the point of loading around the central palatal area. The average dimension was 3 ± 0.2 mm. The width of the load bearing area of the dentures was measured as 42 ± 0.2 mm, and the weight of all denture specimens were measured using an electronic digital scale (Machine Mart Limited, Nottingham, UK). The equivalent flexural strength was calculated in MPa for all denture specimens using Equation (1) [29].
(1)$$\sigma =\frac{3\mathrm{Fl}}{2{\mathrm{bh}}^{2}}$$
where F is the maximum force applied in N, l is the distance between the teeth supports in mm, b is the width of load bearing area of the denture specimen in mm, and h is the thickness of the denture specimen in mm at the point of loading.

Figure 2 presents a picture and a schematic diagram of the bending test experimental setup.

### 2.5. Fracture Behaviour Examination

Following fracture of the denture, the midline fractured surfaces from the bending tests of complete dentures were also studied using a scanning electron microscope (SEM) using a secondary electron detector at an acceleration voltage of 2.0 kV (Carl Zeiss Ltd., 40 VP, Smart SEM, Cambridge, UK) in order to identify the mechanism of failure. Part of the fractured specimens were mounted onto slotted aluminium stubs and coated with a thin layer of gold/palladium using a sputter coater.

### 2.6. Statistical Analysis

The recorded results of bending with and without fatigue loading were calculated and statistically analysed using statistical software (SPSS statistics version 23, IBM, New York, NY, USA). Non-significant Shapiro–Wilk tests demonstrated that data from the bending strength tests was normally distributed and there was homogeneity of variance. A one-way analysis of variance (ANOVA) was used with the Bonferroni significant difference post-hoc test at a preset alpha value of 0.05. In addition, a paired *t*-test analysis was applied to identify any significant difference between the groups at a preset alpha value of 0.05, with and without fatigue loading.

## 3. Results

### 3.1. Weight and Visual Analysis of Denture Specimens

The mean weights of non-reinforced and reinforced complete dentures with 3 wt.% and 5 wt.% zirconia are listed in Table 3. The non-reinforced complete dentures were slightly heavier than the reinforced ones. However, the difference when compared to the reinforced dentures was negligible. This indicated that the addition of zirconia did not significantly change the weight of the dentures.

### 3.2. Cyclic Fatigue Loading

Among the fifteen denture specimens that underwent fatigue cyclic loading tests in the mastication simulator, no dentures failed due to cracking or fracture. This indicated that all dentures, including the reinforced ones, would survive for at least a year in clinical service.

### 3.3. Equivalent Flexural Strength 

Force versus deflection curves of the different denture groups without any fatigue cyclic loading during the bending tests are presented in Figure 3.

The peak breaking forces gradually increased with the increasing percentage of zirconia nanoparticles. Similar behaviour was also noticed for denture specimens following fatigue cyclic loading. One-way analysis of variance (ANOVA) of mean flexure strengths with and without fatigue loading is presented in Table 4.

The specimen groups containing 3 wt.% zirconia with and without fatigue cyclic loading showed a 7.45% and 19.55% increase in the equivalent flexural strength and maximum force, respectively, when compared to the control group. In comparison, the specimen groups containing 5 wt.% zirconia with and without fatigue cyclic loading showed a 10.53% and 26.91% increase in the equivalent flexural strength. The highest increase in the mean value of strength was found for the group containing 5 wt.% zirconia (134.9 MPa) without fatigue cyclic loading, which also showed a significant difference (*p* < 0.05) when compared to the control group (106.3 MPa). However, all the mean strengths of the dentures subjected to fatigue cyclic loading were slightly lower when compared to those of the dentures without any fatigue cyclic loading, but the decrease in mean values were not significant (*p* > 0.05).

### 3.4. Failure Modes of Complete Dentures

After the bending tests with and without fatigue cyclic loading, all 30 denture specimens were examined to identify the failure modes; these are listed in Table 5. The failure modes of the dentures can be broadly classified into two groups: complete fracture and incomplete fracture.

The first general mode of failure is referred to as midline fracture, where the denture was completely broken into two pieces along the midline in the palatal area, as shown in Figure 4. Midline fractures were identified in all groups except the control group with fatigue cyclic loading. The second failure mode can be divided in to two categories: localised fractures that occurred in the area where the load was applied on the denture with the compression head, and cracks that occurred at the free end of the denture. Localised fractures occurred in only 10% of the specimens, which makes it a relatively uncommon failure mode. In contrast, cracks were observed in all specimen groups. This was very common among the failure modes, representing more than 50% of the failures. In addition, no fracture was seen at the anterior and posterior frameworks of the complete dentures.

### 3.5. Fractured Specimen Analysis

Figure 5 shows fractured cross-sections of all three denture specimens (0, 3, 5 wt.% of zirconia) at the point of loading during the bending tests. All surfaces can be characterised by a pattern of globular-shaped peaks and valleys. It appeared that the globular shapes would match with the peaks and valleys in the opposite surfaces of the two broken pieces. Although the surfaces did not show any large cracks or fractures, evidences of microcracks were present. Further magnified views of the surfaces at 1000× revealed patches of smooth surfaces along with rough surface areas. Figure 6 shows the characteristics of the rough and smooth surface areas at high magnification. Small voids were visible in all surfaces. The presence of zirconia nanoparticles was observed in the nanocomposites, particularly in the smooth surface regions with indication of not fully homogeneous distribution. There was also evidence of zirconia particle clustering to a small degree, indicated by circles in Figure 6.

## 4. Discussion

This study evaluated the effect of fatigue cyclic loading on the equivalent flexural strength of complete (maxillary) dentures. The experimental data marginally supported part of the hypothesis of the study that without any fatigue cyclic loading, the equivalent flexural strength of dentures manufactured of nanocomposite with only 5 wt.% zirconia was significantly higher than the control group, but not significantly different from that with 3 wt.% zirconia. However, the other part of the hypothesis, i.e., improvement in flexural strength of PMMA–zirconia denture base with zirconia particles after fatigue loading, was totally rejected.

The flexural strength of denture base materials is generally evaluated by a three-point bending test on a beam shape sample according to the British standard BS 20795-1:2013 (ISO 20795) using a beam shape sample according to Equation (1) [29]. In this study, bending tests were conducted on real dentures in order to conduct experiments to as close a “real” situation as possible. Additionally, using the equivalent flexural strength calculation whilst conducting bending tests directly on dentures could be considered more clinically relevant than standardised tests.

In this study, the increase in equivalent flexural strength with 3 and 5 wt.% of zirconia particles in HI-PMMA could be related to the incorporation of nanoparticles with a size ranging from 30 to 100 nm, which is demonstrably smaller than HI-PMMA powder particles (50 µm). The nanoparticles provide increased surface area to create stronger bonds between the acrylic matrix and the particles. However, the proportion of zirconia nanoparticles should be kept as relatively low as possible to ensure that they can be easily and uniformly embedded within the matrix resin without any significant particle clustering [4].

The surface of the hydrophobic polymer matrix does not wet or react well with the hydrophilic inorganic nanofillers as result of the difference in surface energies [4]. In order to improve wetting of the surfaces and adhesion bonding between the filler and matrix, the surface of hydrophobic fillers should be modified [4]. According to previous studies, the application of silane treatment could play a major role in improving chemical bonds between fillers and polymer matrix, which could therefore increase fracture resistance [3,11]. In this study, the surface of the zirconia nanoparticles was treated with a silane coupling agent that resulted in a strong adhesion between the surfaces of zirconia nanoparticles and PMMA matrix, thus leading to an improvement in the equivalent flexural strength of the nanocomposites [2].

Furthermore, improved particle homogeneity in the HI-MMA liquid zirconia nanoparticle mixture was ensured using a speed mixer machine, which was also thought to contribute to the improvement in the equivalent flexural strength. It is expected that a homogeneous distribution of zirconia particles would fill the spaces between linear chains of acrylic resin matrix. This would therefore restrict the segmental movements of the macromolecular chains and thus improve the flexural strength of the nanocomposite [25].

After the application of fatigue loading cycles, the hypothesis that the nanocomposite dentures would display no statistically significant difference in equivalent flexural strength compared to the control group was rejected. However, 3 and 5 wt.% zirconia-impregnated PMMA dentures showed a slight increase in equivalent flexural strength with fatigue cyclic loading. This implies that under clinical conditions, the nanocomposite dentures would be either as good as, or better than, the control group. A limitation of the study was that small number of specimens for each group was tested. A larger group size would help in distinguishing the difference between them more clearly.

After the fatigue cyclic loading in the mastication simulator for 250,000 cycles, denture specimens did not show any visible cracks or fracture failures, which was equivalent to a patient using a complete denture for approximately one year. A mastication force of 40 N applied on the occlusal surface to each side of the premolar during simulation was similar to the chewing force on one side of the maxillary or mandibular complete dentures worn by a patient as reported in the literature [10]. Similar results were also found in the literature where the performance of acrylic resin denture reinforced with glass fibres and metal mesh was evaluated under a fatigue loading of 80 N and 300,000 cycles. They concluded that the fatigue loading cycles might be insufficient to cause fatigue failure of the dentures [10]. This demonstrated agreement with this current study that no failure occurred after one year of fatigue loading cycles. However, this could be the reason for a decrease in flexural strength for all groups subjected to fatigue cyclic loading compared to that without fatigue cyclic loading. Generally, the fatigue strength of most materials decreased as a result of cyclic stress over a long period of time [7].

The classification of failure modes in this study was based on the location and propagation of fracture lines or cracks in the dentures from the point of loading or stress concentration at the palatal area. Only three types of failure were observed (midline fracture, crack, and localised fracture) unlike the failures mentioned in the literature such as complete tooth failure and denture flange failure [3]. In this study, both with and without fatigue cyclic loading, the fracture in the nanocomposite dentures started near the labial frenum and propagated either between the central, lateral, and canine teeth or first premolar teeth from the polished surface toward the fitting surface, until it reached the loading point, thus resulting in a complete midline fracture. By comparison, one denture from the control group without fatigue cyclic loading showed a midline complete fracture. It is interesting to note that complete midline fractures occurred more frequently in the nanocomposite dentures than the control group. This could be explained by the fact that even though the addition of zirconia in PMMA could increase the equivalent flexural strength, it can, at the same time, also increase the overall brittleness of the denture.

The midline fracture might have occurred as result of the notch shape of the labial frenum, which is considered a potential weak point in the denture structure [11]. Kelly et al. suggested that resistance to fatigue failure of dentures could be improved by eliminating contrasting surface contours such as deep notches at low frenal attachments during manufacture. Furthermore, acrylic resin should be carefully handled during denture fabrication as to avoid any contamination that could influence the presence of localised stress point [6].

The SEM images of the denture specimens were also analysed after the bending tests without conducting any fatigue cyclic loading in the mastication simulation machine. No noticeable differences were observed in the failure mechanism of the dentures with and without the application of fatigue cyclic loading. Only the fractured surfaces from the dentures with fatigue cyclic loading are shown here to represent the worst-case scenario. It was also observed that the size of voids was of the same order as the size of the zirconia particles. Therefore, the particles would presumably fill the empty spaces (Figure 6) and positively affect the strength of the denture. The SEM images also showed that the zirconia particles were fairly distributed within the PMMA matrix without observable particle clustering.

With zirconia-impregnated PMMA, the processing and manufacture of dentures for clinical application would avoid the issues faced with fibre or mesh reinforced dentures, such as longer processing times, incorrect positioning of the fibres within the denture, non-uniform distribution of fibres within the matrix, poor wetting of fibres across the smallest denture thickness and poor bonding between the fibres and the matrix due to lack of polymerisation [24].

## 5. Clinical Implications

This study suggested that maxillary complete removable dentures made of PMMA incorporating a small percentage (5 wt.%) of zirconia nanoparticles could additionally improve the equivalent flexural strength when compared to pure PMMA but not clinically significant under the condition of fatigue loading during mastication.

## 6. Conclusions

Removable complete dentures were made of high-impact (HI) heat-polymerised PMMA resin as a control group and HI-PMMA reinforced with zirconia nanoparticles (3 and 5 wt.%) in order to compare their equivalent flexural strengths with and without applying fatigue loading. Higher equivalent flexural strengths were found for the specimens with 5 wt.% zirconia when the compared with that of 3 wt.% zirconia and the control group, only in cases without fatigue loading cycles. The specimens subjected to fatigue cyclic loading showed an observable decrease in the equivalent flexural strength, but these were not statistically significant when compared to the specimens without fatigue cyclic loading. Within the limitations of this study, it can be concluded that dentures made with zirconia-impregnated PMMA do not result in any significant improvements for clinical application. The common failure modes in the dentures under bending were found to be midline fracture, localised fracture, and cracking. Uniform distribution of zirconia particles was observed in the fractured specimens.

## Figures and Tables

**Figure 1 materials-13-02580-f001:**
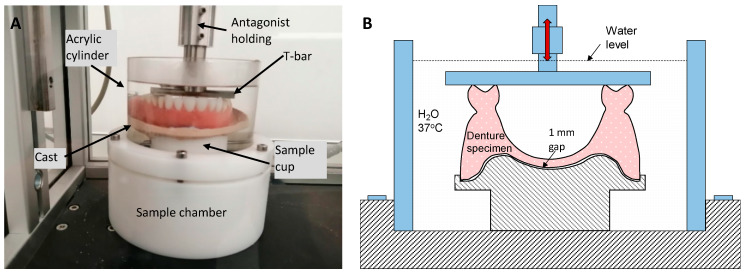
(**A**) Test setup for mastication simulation under fatigue cyclic loading and (**B**) schematic diagram of mastication fatigue loading.

**Figure 2 materials-13-02580-f002:**
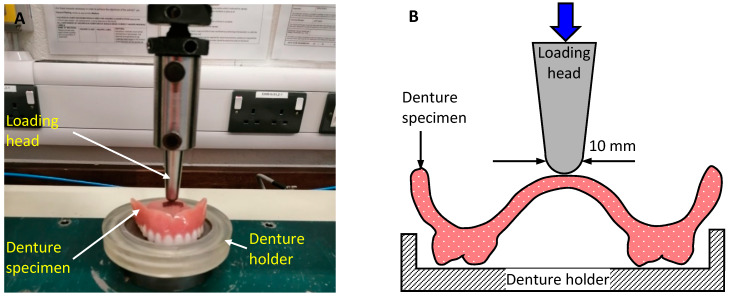
(**A**) Applying bending load on palatal surface of denture specimen in Hounsfield universal testing machine and (**B**) schematic diagram of loading conditions.

**Figure 3 materials-13-02580-f003:**
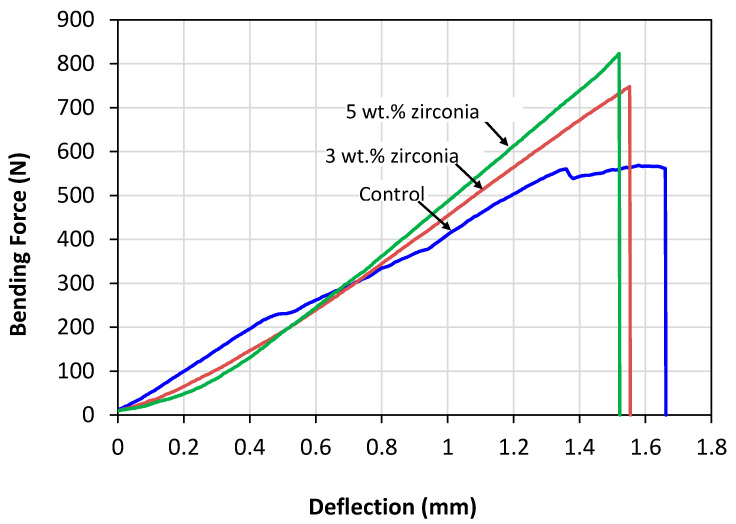
Typical bending load vs. deflection curves without fatigue cyclic loading for pure high impact (HI)-PMMA and zirconia-reinforced nanocomposites.

**Figure 4 materials-13-02580-f004:**
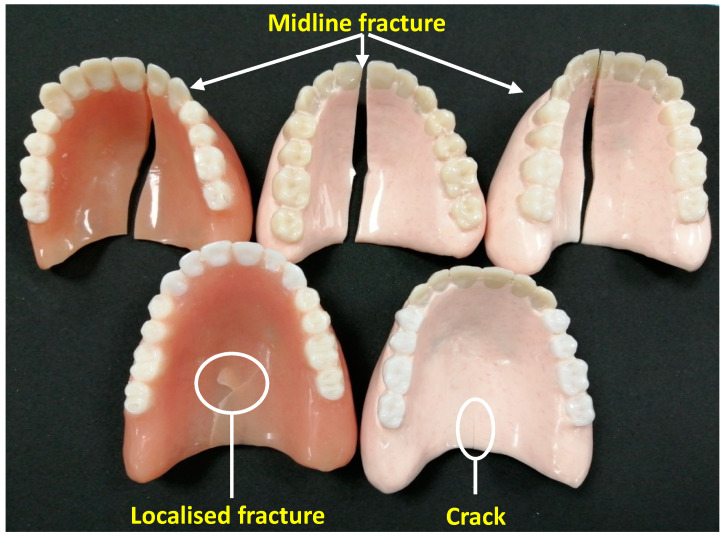
Failure modes observed in the dentures during the bending tests: midline fracture, localised fracture, and crack.

**Figure 5 materials-13-02580-f005:**
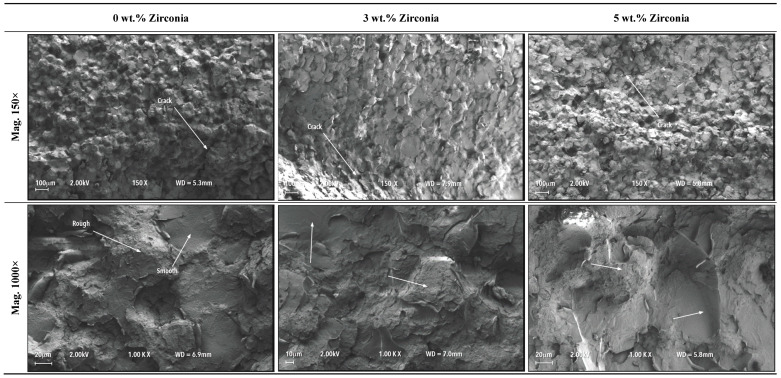
Fractured surfaces of denture specimens with fatigue cyclic loading during bending tests at different magnifications.

**Figure 6 materials-13-02580-f006:**
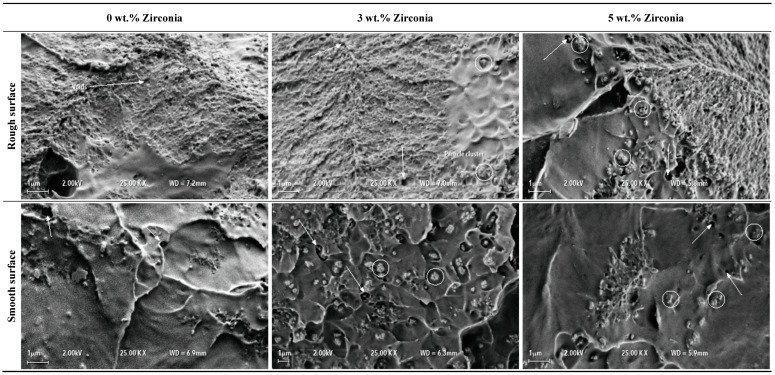
Fractured surfaces of dentures with fatigue cyclic loading during bending tests showing surface characteristics at 25,000×. Arrows indicate voids and circles indicate voids filled by zirconia particles.

**Table 1 materials-13-02580-t001:** Materials used in making complete removable dentures.

Materials	Trade Name	Manufacturer	Lot. Number
High-impact heat-curing acrylic denture base resin	HI Metrocryl	Metrodent Limited, Huddersfield, UK	Powder (22828) Liquid (103/4)
Yttria-stabilised zirconium oxide	Zirconium oxide	Sky Spring Nano Materials, Inc., Houston, TX, USA	8522–120315
Dental plaster	Flasking plaster	Saint-Gobain, Formula, Newark, UK	0411217–3
High-strength dental stone	Dentstone KD	Saint-Gobain, Formula, Newark, UK	085217–5
Type 4 diestone	Metrostone	Metrodent Limited, Huddersfield, UK	032218–1
Addition cure silicone putty 1:1	Sheraduplica	Shera, Lemforde, Huddersfield, UK	Base (86392) Catalyst (86047)
3-Trimethoxysilyl propyl methacrylate	Silane coupling agent	Sigma Aldrich, Gillingham, UK	440159

**Table 2 materials-13-02580-t002:** Weight percent zirconia in combination with acrylic resin powder as well as monomer content of the specimen groups.

Experimental Groups	Zirconia (wt.%)	Zirconia (g)	HI PMMA Powder (g)	HI MMA Monomer (mL)
Control	0.0	0.000	21.000	10.0
Nanocomposite-1	3.0	0.630	20.370	10.0
Nanocomposite-2	5.0	1.050	19.950	10.0

**Table 3 materials-13-02580-t003:** Weight of complete dentures made of pure polymethyl methacrylate (PMMA) and zirconia-impregnated PMMA.

Weight of Non-Reinforced PMMA Dentures (g) (Mean ± SD)	Weight of Reinforced PMMA Dentures (g) (Mean ± SD)	
Control Group 0 wt.% of zirconia	3 wt.% of zirconia	5 wt.% of zirconia
20.1 ± 1.0	19.5 ± 0.2	19.5 ± 1.0

**Table 4 materials-13-02580-t004:** Maximum force (N) and mean and SD of values the equivalent flexural strength (MPa) before and after fatigue cyclic loading for the test groups.

	Without Fatigue Cyclic Loading		With Fatigue Cyclic Loading	
Weight Percent Zirconia	Maximum Force (N)	Equivalent Flexural Strength (MPa) and SD	Maximum Force (N)	Equivalent Flexural Strength (MPa) and SD
Control (0.0 %)	633.2	106.3 (21.3)^Aa^	598.9	100.6 (17.4)^Aa^
3.0 %	757.0	127.1 (5.8)^Ab^	643.8	108.1 (15.2)^Ab^
5.0 %	803.6	134.9 (13.9)^Bc^	662.2	111.2 (15.45)^Ac^

Note: Within a column, cells having similar (upper case) letters are not significantly different from the control group (0% zirconia content) and within a row values identified using the same lower-case letters are not significantly different; n = 5 specimens per group.

**Table 5 materials-13-02580-t005:** Failure modes of complete dentures with and without fatigue cyclic loading.

Failure Mode	Name of Failure Modes	Control Group (0 wt.% Zirconia)		3 wt.% Zirconia		5 wt.% Zirconia	
		Without Fatigue Loading	With Fatigue Loading	Without Fatigue Loading	With Fatigue Loading	Without Fatigue Loading	With Fatigue Loading
Complete fracture	Midline fracture	1	0	2	2	3	3
		Between central and lateral	-	One between central and lateral. One between centrals	One between canine and first premolar. One between central and lateral	Two between centrals. One through a central	One between centrals. One through a lateral. One between central and lateral
Incomplete fracture	Localised fracture	1	1	1	0	0	0
	Cracks	3	4	2	3	2	2

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
