# Peer review of "Evaluation of Equivalent Flexural Strength for Complete Removable Dentures Made of Zirconia-Impregnated PMMA Nanocomposites"

_materials, 2020, doi:10.3390/ma13112580_

Round 1

Reviewer 1 Report

 “The aim of this study was to investigate the flexural strength of complete removable dentures made of HI heat polymerized PMMA resin, reinforced with zirconia nanoparticles at two different concentrations. The effect of fatigue loading on the flexural strength behaviour of the dentures was also investigated.”

General remarks

The topic is interesting and important to warrant publication. The paper is written well, properly organized, and easy to follow. However, the conclusions and the clinical implications must be corrected:

The study showed that denture specimens made of HI heat polymerized PMMA reinforced with 5 wt % zirconia nanoparticles significantly improved equivalent flexural strength compared to that made of pure PMMA when the specimens were not subjected to any prior fatigue cyclic loading. When the dentures were subjected to fatigue cyclic loading, the decrease in mean values were slightly lower when compared to those of the dentures without any fatigue cyclic loading, but with no significant difference. There was also no significant difference between the control group and the 3 wt % and the 5 wt % groups, after fatigue cyclic loading.

Since studies in- vitro should always mimic mouth conditions, the results comparing the groups are more clinically relevant after the specimens went through fatigue cyclic loading. In the current study, the results between the groups were not statistically different after fatigue cyclic loading.

Moreover, the failure mode showed very dramatically that complete midline fractures occurred more frequently in the nanocomposite dentures than in the control group. The explanation can be, as was written in the discussion, that the addition of zirconia in PMMA could increase the overall brittleness of the denture. Therefore, the study hypothesis should be totally rejected and the conclusion and the clinical implications must be corrected.

A limitation of the study is the small amount of specimens for each group. This should be mentioned in the discussion. Maybe, larger groups will give a significant difference between them.

Minor revision:

Rewrite the references according to the Journal requirements

Author Response

Hi Dear Reviewer 2

Please see the attachment below of the response comments.

Best wishes

Saleh Zidan

Reviewer 2 Report

Review

Manuscript titled „ Evaluation of Equivalent Flexural Strength for 2 Complete Removable Dentures made of Zirconia 3 Impregnated PMMA Nanocomposites “ is original. The results are interpreted appropriately. The article is written in an appropriate way.

The fact that authors decided to use finished removable dentures instead of samples (that are usually used in artices) is huge advantage of the article. This method of research is the best to present clinical situation.

I have a few advices though:

  1. It is common knowledge that powders tend to aggregate. It is advisable to add the microscope photo of the sample that presents that nanomolecules of ZrO2 are evenly distributed in the sample after mixing it with acrylic resin monomer. In the Figure 9 it is visible that nanomolecules of ZrO2 are not distributed evenly.
  2. How did You count „Maximum force (N) and mean & SD of values the equivalent flexural strength (MPa) before and after fatigue cyclic loading for the test groups” that is presented in Table number 4?
  3. You should put Your attention to removable dentures colour after adding nanomolecules of ZrO It is clinically unacceptable.

Author Response

Hi Dear Reviewer 3

Please see the attachment below of the response comments.

Best wishes

Saleh Zidan

Reviewer 3 Report

Dear Authors,

Here you can my find observations regarding your work:

Abstract: Line 20/21 – It should be clearer as to how many samples were divided respectively to each group (10 each).

Introduction: Line 47. Reference number 2 should be placed at the end of the sentence

Line 52. The statement is not original data or results from the referenced paper (5), but a citation from earlier papers

Line 52-54. The data is from 1981, almost 40 yrs. old, newer data/reference should be used or such statements are irrelevant.

Line 58 – 60. The statement is not original data or results from the referenced paper (3), but a citation from earlier papers Line 66. Reference number (13) should be placed at the end of the sentence.

Line 69. All of the studies were in vitro studies and did not have a clinical (patient) testing component. Should be revised.

Line 70. Reference number 19 does not investigate metal wire reinforcements for acrylic dentures, but only E-glass reinforcements.

Line 73 -79. The statement is not original data or results from the referenced paper (3), but a citation from earlier papers

Line 85. Punctuation missing after “et al”. There is no need to put publication years in brackets after author referencing.

Line 99. Use of three references used for this statement is unnecessary, the references are used only in this sentence and should be reduced to just one (suggestion - Cavalcanti, A.N., et al., Y-TZP ceramics: key concepts for clinical application. Oper Dent, 2009. 34(3): p. 34451.)

Line 104-106. The hypothesis should be rewritten more precisely – “a positive effect” can be also a decrease in value for some situations so, a clear statement “would lead to a significant increase in value of the flexural strength of the complete dentures” should be used.

Materials and Methods: Line 118-120/Table 2/Line 154-157. The group preparations and divisions are scattered across these lines and should be grouped and clearly stated in one place. Also, in line 154 it is not stated who manufactured the specimens (one or more trained experienced
professionals – dental technicians/dentist, if more than one manufacturer has there been an IRR calculated on the thickness of the material etc.).

Line 120-123. This sentence should be moved to the introduction (suggestion before line 99).

Line 139. What was the method of silanization of zirconia?

Line 159. Refence number (9) at the end of the sentence.

Line 199. In the brackets the firm headquarters city, state, county is missing – New York, NY, USA.

Results: Line 213-219/Figure 3. Please explain the values of these findings? It was not elaborated it in the introduction and not stated in the aim of this paper to investigate color, shine, appearance or surface roughness, also no scientific evaluation/testing of these parameters except visual inspection was performed.

Line 224-249. The results of “Equivalent flexural strength” testing are repeated in the text, in Table 4, and Figure 5. Choose which one you prefer (visual – figure, textual – written/table) and state it only once, no need to repeat results. Also, in Table 4 the results of the “before and after” fatigue cycling are not statistically significantly different and not necessary to be shown. Table 4 is difficult to understand because of them and should be removed if possible. The p value set under 0.05 should be stated in the description of the table.

Line 262./Figure 7. The overall percentage and mode of fractures were explained and presented earlier in the text and Figure 6., this statement and Figure 7 are unnecessary and should be removed.

Line 286. Zirconia is not a glass ceramic, and this terminology is incorrect. Zirconia particle, or globes should be used.

Disscusion:

The British standard BS 2487:1989 (ISO 1567) is an old, outdated ISO standard, and has since been revised and replaced by two newer version, the most recent one being the ISO 207951:2013 which should be cited as a refence on the three-point bending test.

“Transformation toughening” is a phenomenon attributed to phase transformation of ZrO2 crystals, but only in solid structures, not hybrid/composite structures with zirconia powder mixed in. Cited studies 2, 23, and 29 (regarding nanoparticles and flexural strength) do no mention “transformation toughening” anywhere in the text and this statement should be removed.

The particles would “presumably” fill the empty spaces, without confirmation it is not definite.

“significant” particle clustering – should be changed to “observable” particle clustering.

Clinical implications

3 wt% incorporation of ZrO2 has been shown by results in this paper not to have a significant difference in equivalent flexural strength, why state that it could improve clinical results? Only 5 wt% showed statistically significant differences.

Author Response

Hi Dear Reviewer 4

Please see the attachment below of the response comments.

Best wishes

Saleh Zidan

Round 2

Reviewer 1 Report

Minor revisions:

Abstract

…”Denture specimens made of HI heat polymerized PMMA reinforced with 5 wt % zirconia nanoparticles significantly improved equivalent flexural strength compared to that made of pure PMMA when the specimens were not subjected to any prior fatigue cyclic loading. In addition, the application of fatigue cyclic loading did not significantly influence the equivalent flexural strengths of all denture specimen groups”

From the abstract, it seems that for clinical use, it is better to add 5 wt % zirconia nanoparticles to the denture. The conclusions in the abstract are not compatible with the conclusions and the clinical implications in the manuscript itself.

Please rewrite the conclusions in your abstract so it will be compatible with your manuscript

And add this sentence to the abstract: “Within the limitations of this study, it can be concluded that dentures made with zirconia impregnated PMMA does not make any significant improvement for clinical application”

Clinical Implication

“This study suggested that maxillary complete removable dentures made of PMMA incorporating a small percentage (3 wt% and 5 wt%) of zirconia nanoparticles could additionally improve the equivalent flexural strength when compared to pure PMMA but not clinically significant under the condition of fatigue loading during mastication.”

Only 5 wt% improves the equivalent flexural strength, not 3 wt%.

Please correct

Conclusions:

….”Higher equivalent flexural strengths were found for the specimens with 5 wt% zirconia when compared with 3 % zirconia and the control group, both with and without fatigue cyclic loading cycles. The specimens subjected to fatigue cyclic loading showed an observable decrease in the equivalent flexural strength, but these were not statistically significant when compared to the specimens without fatigue cyclic loading. Within the limitations of this study, it can be concluded that dentures made with 5% zirconia impregnated PMMA does not make any significant improvement for clinical application. The common failure modes in the dentures under bending were found to be midline fracture, localised fracture and cracking. Uniform distribution of zirconia particles was observed in the fractured specimens”

The first sentence in this paragraph is not correct. “Higher equivalent flexural strengths were found for the specimens with 5 wt% zirconia when compared with 3 % zirconia and the control group, both with and without fatigue cyclic loading cycles.”

Only without fatigue cyclic loading cycles. With fatigue cyclic loading, the 5 wt% had not higher flexural strength. Please correct

Please correct the sentence:

“Within the limitations of this study, it can be concluded that dentures made with zirconia impregnated PMMA does not make any significant improvement for clinical application.”

Author Response

Hi Reviewer 2

Please see the attachment below.

Reviewer 3 Report

Dear Authors, thank you for your revisions.

Author Response

Hi Reviewer 4

Please see the attachment below.
